# Droplet digital PCR-based testing for donor-derived cell-free DNA in transplanted patients as noninvasive marker of allograft health: Methodological aspects

Frederik Banch Clausen[1]*, Kristine Mathilde Clara Lund Jørgensen[1,2], Lasse Witt Wardil[1,2], Leif Kofoed Nielsen[2], Grethe Risum Krog[1]

1 Department of Clinical Immunology, Copenhagen University Hospital, Copenhagen, Denmark,
2 Department of Technology, Faculty of Health, University College Copenhagen, Copenhagen, Denmark

☉ These authors contributed equally to this work.
* frederik.banch.clausen@regionh.dk

## Abstract

In solid organ transplantation, donor-derived cell-free DNA (dd-cfDNA) is a promising universal noninvasive biomarker for allograft health, where high levels of dd-cfDNA indicate organ damage. Using Droplet Digital PCR (ddPCR), we aimed to develop an assay setup for monitoring organ health. We aimed to identify the least distinguishable percentage-point increase in the fraction of minute amounts of cfDNA in a large cfDNA background by using assays targeting single nucleotide polymorphisms (SNPs). We mimicked a clinical sample from a recipient in a number of spike-in experiments, where cfDNA from healthy volunteers were mixed. A total of 40 assays were tested and approved by qPCR and ddPCR. Limit of detection (LOD) was demonstrated to be approximately 3 copies per reaction, observed at a fraction of 0.002%, and which would equal 6 copies per mL plasma. Limit of quantification (LOQ) was 35 copies per reaction, estimated to 0.038%. The lowest detectable increase in percentage point of dd-cfDNA was approximately 0.04%. Our results demonstrated that ddPCR has great sensitivity, high precision, and exceptional ability to quantify low levels of cfDNA. The ability to distinguish small differences in mimicking dd-cfDNA was far beyond the desired capability. While these methodological data are promising, further prospective studies are needed to determine the clinical utility of the proposed method.

## Introduction

When patients receive a solid organ transplantation, efficient monitoring of the allograft health is important for organ survival [1]. Therefore, organ recipients must regularly undergo a tissue biopsy to monitor the allograft health. Such current method is an invasive diagnostic procedure that may cause discomfort and rare but serious complications [2, 3]. Some noninvasive markers exist [3], but the predictive value is either limited or unreliable [4, 5]. One promising alternative is donor-derived cell-free DNA (dd-cfDNA) [1, 3, 6].

**Data Availability Statement:** All relevant data are within the paper and its Supporting information files.

**Funding:** The authors have no competing or financial interests regarding the submitted manuscript. FBC received funding from The Toyota-Foundation (KJ/BG-9881 H), The Blood Donors' Research Foundation (J.no. FF 4 2021/01) (www.blodonor.dk/fonde), and Doctor Sofus Carl Emil Friis and Wife Olga Doris Friis' Foundation (11062020). The funders had no role in the study design, data collection and analysis, decision to publish, or preparation of the manuscript.

**Competing interests:** The authors have declared that no competing interests exist.

Transplanted organs release cfDNA into the recipient's bloodstream, and this dd-cfDNA is detectable in an ordinary blood sample taken from the patient [1, 2, 7]. Analyzing dd-cfDNA has shown great potential as a biomarker for monitoring the allograft health, where an increase of the dd-cfDNA fraction in the recipient's bloodstream will indicate organ damage [2, 8–12]. As a noninvasive marker for organ health, dd-cfDNA testing has been applied for various organs [3, 6], including the heart [13–22], kidney [5, 23–29], liver [11, 29], lung [30–32], and pancreas and simultaneously pancreas-kidney transplantation [33].

There are numerous, potential benefits from using dd-cfDNA testing. Monitoring allograft health based on a blood sample would be less invasive and cause the patient less discomfort compared to a regular tissue biopsy [34, 35]. dd-cfDNA-based monitoring may predict early rejection as well as subclinical damage, and continuous dd-cfDNA-based allograft health monitoring may allow for a better adjusted immunosuppressive therapy [36]; the potential effect of treatment may also be monitored by dd-cfDNA testing. dd-cfDNA has repeatedly been shown to provide an early indication of organ damage, thus unmasking pathology long before existing tools. The ultimate clinical benefit would be prolonged organ survival and thus prolonged patient survival.

Methodologically, two overall approaches are used, which are based on either random or targeted differentiation of donor and recipient cfDNA using different methods of either next generation sequencing (NGS), Droplet Digital PCR (ddPCR), or quantitative real-time PCR (qPCR) [1, 8]. Various commercial solutions are available, including TRAC, TheraSure, Allo-Sure and Prospera [8].

Beck and colleagues developed a ddPCR-based method capable of monitoring dd-cfDNA, first presented in 2013 [9]. Beck et al. use a selection of assays that specifically target biallelic, single nucleotide polymorphisms (SNPs) to differentiate between donor and the recipient cfDNA in a plasma sample. In each assay, two probes are used, each probe detecting each SNP allele. A given biallelic SNP assay can distinguish between donor and recipient when donor and recipient are homozygous for each different SNP allele [9, 10, 37]. Statistically, this situation will occur for 12.5% of the SNPs (assuming an allele frequency of 0.5 for both alleles in a biallelic SNP target).

In ddPCR, each reaction mixture is divided into separate PCR reactions in up to 20,000 droplets. This partitioning of target molecules in droplets allows for a high-precision quantification method (without the use of standard curves)—creating a quantitative precision which is much greater than the precision of qPCR, especially in cases of low levels of DNA [34, 35, 38]. ddPCR-based analysis of dd-cfDNA has been successfully demonstrated for all organs, including kidney [27, 39]; heart [40], lung [31], and liver [11].

When an allograft is in a stable phase, the fraction of dd-cfDNA compared to the recipient's total cfDNA has been reported to be approximately 10% for liver transplants, 1% for kidney transplants, and 0.5% for heart transplants [9, 10]. Consequently, it is important that the measuring method is highly sensitive as well as capable of providing precise quantification at low DNA levels. Importantly, high precision is necessary to distinguish a small increase or decrease from stochastic or biological variation. In addition, a robust algorithm must be established for when organ damage can be indicated. The specific levels of dd-cfDNA for each organ at steady state, the specific algorithm for predicting subclinical organ damage, as well as the extent of clinical actions needed or taken are all essential elements that are not yet fully established.

In this study, we assessed various key analytical performance parameters of ddPCR. We developed an arsenal of SNP assays, partly based on the design from Beck and colleagues and partly on our own design. We investigated the limit of detection (LOD), the limit of quantification (LOQ), and we aimed to identify the lowest detectable increase in fractions measured in

percentage points. This was done by quantifying minute concentrations of cfDNA in experiments using samples spiked with cfDNA.

## Materials and methods

### Ethics statement

This study was a quality assurance project using blood samples from healthy volunteers. Informed and written consent was obtained from all participating volunteers when blood samples were collected. According to Danish law, no ethical approval was required, because this was a quality assurance project, thus waiving the need for ethics committee approval of the study. No DNA information related to disease was examined, and only SNPs with no known clinical significance were used, thus avoiding the challenges of reporting incidental findings.

### Sample material

We sampled blood from a total of 32 healthy volunteers for assay development and for spike-in experiments designed to mimic dd-cfDNA in transplanted patients. No patients with organ transplantations were tested in this study. All blood samples were collected and analyzed from 2020 to 2022 in The Laboratory of Blood Genetics, at The Department of Clinical Immunology, Copenhagen University Hospital, Copenhagen, Denmark.

### Sample collection and preparation

Blood samples collected from all included volunteers were genotyped for the selected SNPs. All blood samples were collected in 10 mL Cell-Free DNA BCT tubes (Streck, NE, USA). Within one hour after blood collection, plasma was separated from cells by centrifugation for 10 min at 1990x*g* at room temperature. cfDNA was extracted from 4 mL plasma using the automated QIAsymphony SP extraction system (Qiagen, Hilden, Germany) using the QIAsymphony DSP Virus/Pathogen-kit (Qiagen). Extracted cfDNA was eluted in 60 μL AVE buffer. Eluted cfDNA was stored at −20˚C until further use.

Samples were picked for the spike-in experiments based on their genotype where cfDNA from one individual (homozygous for one SNP allele) would be added to cfDNA from another individual (homozygous for the other SNP allele for a given SNP assay). For the sake of clarity, we will henceforth refer to spike-in material as mimicking dd-cfDNA and the donated background DNA as background cfDNA.

### SNP assay selection

Each biallelic SNP assay consisted of a forward and reverse primer, and two probes targeting each of the biallelic SNPs. The two probes were labelled with either a FAM or a HEX fluorescent dye, in conjunction with a Black Hole Quencher 1 (BHQ1) (Eurofins MWG, Edersberg, Germany). SNP assays were adopted from Beck and colleagues [37] and analyzed for their suitability in our laboratory setting. We did *in-silico* verification and quality assurance of all SNP assays which included checking correct DNA sequences, potential overlap of primers and probes, melting temperatures, and potential secondary structure formations. If sequences overlapped, or did not match blasted sequences, or other criteria was not met, the assay was discarded. Assays which were approved by *in silico* verification were then subjected to qPCR and ddPCR testing, as described below.

22 out of 40 assays described by Beck et al. did not pass our quality assurance. We therefore designed new assays. In short, we designed assays for non-clinical, biallelic SNP targets with a minor allele frequency of 0.4 to 0.5 and with no additional SNPs of >1% frequency present in

the primer or probe sequences, and with a maximum amplicon length of 90 bp. Please see full description of design criteria in S1 Text.

Each SNP assay was thoroughly tested by qPCR to check for the capability of allele discrimination using all 32 control individuals as material to allow assessment of acceptable allele discrimination. qPCR was done using qPCR 7500 Fast Real-Time PCR Systems (Applied Biosystems) and by using the 2x Universal Master Mix with uracil-N-glycosylase (UNG) (Applied BioSystems), primers and probes in the same concentration as used in ddPCR, and 2–10 ng genomic DNA in a total reaction volume of 10 μL. The thermal profile was common to all assays consisting of 1 min at 60˚C and 10 min at 95˚C, followed by 40 cycles of 95˚C for 15 sec and 60˚C for 1 min with a post-PCR read at 60˚C for 1 min. If more than tree clusters were formed in the allelic discrimination plot or allele calling was not automatic with default settings, assays were discarded. Subsequently, discrimination capability was controlled with ddPCR using homozygous DNA for each variation in the biallelic SNP and heterozygous DNA.

A number of SNP assays were selected (at random) for assessing the coefficient of variation (CV%) of measurements of 0.5% fraction at varying concentrations and for a deeper assessment of LOD, LOQ, and minimum distinguishable percentage point difference, as assessed by spike-in experiments.

## Preamplification of cfDNA

We used preamplification to make sufficient DNA material for the spike-in experiments. Extracted plasma cfDNA was preamplified with 12 PCR cycles when acting as mimicking dd-cfDNA and 20 PCR cycles when acting as background cfDNA. The expected final concentration was calculated using an equation from Andersson et al. [41]. Subsequently, a ddPCR run was made to measure the exact DNA concentration of the pre-amplified DNA (using 20x dilutions or more if more than 5000 copies/μL was expected), following which dilutions for spike-in experiments were made.

Preamplification was done in a 25 μL reaction volume consisting of 2x ddPCR Supermix for probes (No dUTP) (Bio-Rad), primers used in the specific SNP assay at a final concentration of 40 nM, 10 μL cfDNA, topped up with sterile $H_2O$ ($sH_2O$). Amplification was done with a Veriti 96 Well Thermal cycler (Applied Biosystems) using the following PCR conditions: 95˚C for 10 min initial denaturation, 12 or 20x (94˚C denaturation for 30 s, 60˚C annealing and extension for 1 min) then a final extension at 72˚C for 10 min. Two no template controls (NTC) were analyzed on par with the amplified cfDNA with every SNP assay.

## Droplet digital pcr

We used the QX200 ddPCR system (Bio-Rad, Hercules, California, US) to determine the concentration of cfDNA. Each reaction with a final volume of 20 μL consisted of 2x ddPCR Supermix for probes (No dUTP) (Bio-Rad), primers and probes at a final concentration of 900nM and 250nM respectively, 8 μL cfDNA topped up with sH2O. Droplets were generated using the QX200 Droplet Generator (Bio-Rad) and subjected to DNA amplification on the C1000 Touch Thermal Cycler (Bio-Rad) with the following PCR conditions: 95˚C for 10 min initial denaturation, 40x (94˚C denaturation for 30 s, 60˚C annealing and extension for 1 min), then 98˚C for 10 min for enzyme deactivation. The ramp rate was 2˚C/s. The ddPCR plate was held at 4˚C until further analysis. Droplets were read by the QX200 Droplet Reader (Bio-Rad) and analyzed using QX Manager Software Standard edition version 1.2.345.

We used Direct Quantification (DQ), and a threshold value was set manually for each SNP assay before interpreting the results.

The QX Manager Software displayed all measured DNA concentrations in copies/μL and in copies per 20 μL reaction mix (henceforth referred to as copies/reaction). (A given DNA concentration of 5 copies/μL is thus equivalent to 100 copies/reaction; for a clinical plasma sample, this DNA concentration would further correspond to an initial cfDNA concentration of 750 extractable copies per 4 mL plasma [copies per reaction / template volume * elution volume = 100 / 8 * 60 = 750 copies]).

## Limit of detection

In an experiment (using SNP assay S68), we analyzed thirteen different concentrations of mimicking dd-cfDNA from 0.15 to 400 copies per μL spiked into a constant background of approximately 6500 copies per μL of background cfDNA, equivalent to a fraction range of 0.002–6%. The specific DNA concentrations were 400, 200, 100, 50, 25, 15, 10, 5, 2.5, 1.25, 0.63, 0.3, and 0.15 copies per μL reaction mix. Each DNA concentration was analyzed in duplicate.

LOD was defined as the lowest concentration detected. The dynamic range is described as 1–120,000 copies/reaction by the manufacturers. The data from the LOD experiment were subjected to adjustment; the background signal from any unspecific reaction from the opposite allele (false positive signal, FPS) was subtracted from all the measured spike-in concentrations, as recommended by Kokelj et al. [39]. In this experiment, the FPS was 0.13 copies per μL.

## Quantitative experiments

To assess quantitative capabilities, five spike-in experiments were performed, henceforth referred to as Spike 1–5. In Spike 1 and 2 (using assay S226), we analyzed a fixed fraction of 0.5% (as a first representative level of dd-cfDNA) using different absolute DNA concentrations. The following solutions were tested: 100 copies/reaction in a background of 20,000 copies/reaction; 200 copies in 40,000 copies, and 300 copies in 60,000 copies.

In Spike 3–5, we further investigated the detection of lower levels of DNA to estimate LOQ and least distinguishable change in fraction. In Spike 3–5 (using SNP assay S67, S68, and S110, respectively), the background cfDNA concentration was held constant while the concentration of the mimicking dd-cfDNA varied in four different concentrations (with varying distances between values), as described below. For Spike 3, the mimicking dd-cfDNA was diluted to concentrations of 12.5, 25, 50, and 100 copies/reaction, tested in a final background cfDNA concentration of 94,000 copies/reaction. For Spike 4, the mimicking dd-cfDNA was diluted to 12.5, 25, 100, and 200 copies/reaction, tested in a background cfDNA concentration of 47,000 copies/reaction. For Spike 5, the mimicking dd-cfDNA was diluted to 34, 68, 137, and 275 copies/reaction, tested in a background cfDNA concentration of 94,000 copies/reaction. The highest mimicking dd-cfDNA concentration was tested in 12 replicates, and the lower concentrations were tested in 18 replicates. A full 96-well ddPCR plate was used per spike-in experiment, with two NTCs per column. In the first column of each ddPCR plate, dilutions of the background cfDNA concentration and the mimicking dd-cfDNA were analyzed in triplicates. The mean concentrations of these measurements were then used to determine the exact, expected concentration of the mimicking dd-cfDNA concentrations in each spike-in experiment, as presented in the results.

FPS from the opposite allele was subtracted from all the measured mimicking dd-cfDNA concentrations. The FPS was 4.2 copies/reaction for Spike 3, 1.0 copies/reaction for Spike 4, and 9.6 copies/reaction for Spike 5. We chose an acceptable CV at <25% as the criterion defining the LOQ.

We calculated a 95% confidence interval of the fractions for each spike-in experiment. To distinguish between two fractions, we defined that the confidence intervals of each fraction should be non-overlapping. For Spike 3–5, we identified the lowest detectable increase in fractions between mean fraction values for each experiment, measured as difference in percentage points. Percentage points were calculated by subtracting the low mean fraction value from the higher mean fraction value where the 95% confidence intervals did not overlap each other.

### Statistics

Data were exported from the QX Manager Software to Microsoft Excel 365 where mean concentrations, standard deviation (SD), coefficient of variation (CV), mimicking dd-cfDNA fractions, and 95% confidence intervals for the fractions were calculated. Linear regression was used to evaluate linearity, and Pearson's r was used to evaluate the precision between the measured and expected concentrations of the mimicking dd-cfDNA. Measurements of 0.5% fractions (for Spike 1 and 2) were evaluated using ANOVA (Tukey's multiple comparisons test) using Graph Pad Prism 9.

### Results

We validated 40 SNP assays for ddPCR-based cfDNA quantification to enable high-precision quantification of dd-cfDNA in transplanted patients. All assays are presented in Table 1. The average amplicon length of each assay was 85.4 bp (range: 66–103 bp). We based our setup on the design from Beck and colleagues [37], of which 18 assays were approved and selected, and 22 were discarded because they did not meet our criteria for inclusion. Therefore, we designed 22 new assays; average amplicon length was 82.3 bp (range: 66–90 bp).

We investigated the LOD using dilutions of cfDNA, and the lowest DNA concentrations studied were easily detected by the ddPCR method. Thus, we demonstrated a LOD of 3 copies/reaction equivalent to a fraction of 0.002%, which would theoretically be equivalent to 6 copies per mL plasma in our setup using DNA extraction from 4mL plasma. Excellent linearity was demonstrated with high correlation between expected and measured absolute values (Pearson's $r = 0.997$ [95%CI = 0.989–0.999]), Fig 1A. Similar, excellent linearity was shown for fractions of mimicking dd-cfDNA (Pearson's $r = 0.996$ [95%CI = 0.986–0.999]), Fig 1B.

We then investigated quantitative capabilities in spike-in experiments, designated Spike 1–5. In Spike 1–2, we examined a repeated detection of a 0.5% fraction, Fig 2. A similar detection of 0.5% was observed in all three dilutions with different absolute values ranging from 100–300 copies/reaction (0.52%, 0.50%; and 0.45%, respectively, in Spike 1); there were no statistically significant differences between the different 0.5% measurements, except from the difference between 100 and 300 copies/reaction in Spike 1 ($p = 0.0469$). In Spike 1, CV was 10.9%, 9.3%, and 9.1% for 100–300 copies/reaction, respectively. In Spike 2, the detection was 0.50%, 0.50%; and 0.51%, respectively; CV was 13.6%, 7.4%, 7.7%, respectively.

In Spike 3–5, we then investigated a repeated detection of lower DNA concentrations to determine LOQ, and we studied the different fractions of mimicking dd-cfDNA to determine the lowest detectable percentage-point increase between low levels of mimicking dd-cfDNA. First, we compared the expected concentration of the mimicking dd-cfDNA with the measured concentration to evaluate our dilutions as well as the measurement precision in the lower part of the ddPCR method's dynamic range. The results demonstrated excellent agreement between expected and measured values over the range of DNA concentrations tested, Fig 3, with Pearson's $r = 0.9987$ for Spike 3, $r = 0.9998$ for Spike 4, and $r = 0.9998$ for Spike 5.

The expected and measured DNA concentrations are presented in Table 2, along with CV values and fractions for each experiment. Overall, the measured concentration of the

**Table 1. Overview of SNP assays.**

| Assay | SNP ID | Alleles | Chr:position | Global MAF | Forward Primer | Reverse Primer | Probe A (5'-FAM/3'-BHQ1) | Probe B (5'-HEX/3'-BHQ1) | Amplicon (bp) |
|---|---|---|---|---|---|---|---|---|---|
| S43 | rs741384 | C;G | 2:216687231 | 0.49/0.51 | GTCTCTGGGGGTCTGTTGGCC | AGAGGAAGGACTCCCAGGGGG | TGGAGACGGGTCCGCAGAG | TGGCACAGGTGCTCTCCGG | 100 |
| S48 | rs13333451 | C;G | 16:87672262 | 0.48/0.52 | GATCAACTCCTGAAGAGACTCCGT | AGGGAGGGATGAGAGAGGGAC | CGGGAGCCCTGCGCTTTG | TTTCCATGACAAACCGCAGGG | 91 |
| S55 | rs3909244 | C;G | 18:3769560 | 0.49/0.51 | TGGTTAAACTGTAGTACATCCATGGA | ACCTTTTGGGACTGGCTTTCT | ACTTTCTCAGCAACAGCTGA | CCTGGAAATTCATCCAGCCTGT | 98 |
| S58 | rs1317808 | C;G | 2:47627849 | 0.54/0.46 | GCCACCTTAGCCTCCCAAAG | AGGGTGACTGTATTATTATTGTTCAAACT | ATTACAGGCATGAGCCACCG | CAAGGCACGGTGCCTCAT | 87 |
| S63 | rs2312356 | C;G | 7:153385741 | 0.47/0.53 | GCTGTTGCTGCCTCACAGGT | AGGGCAAAGCAAATGCACCA | AACTGGAAGTAACACCTGCACCA | CTTGACTCTTGGTGCACGTGT | 103 |
| S66 | rs1522662 | G;C | 11:81114448 | 0.48/0.52 | ACCCTGACCCTCAGTTCCTT | AAGAGCCCTTATAAGGTGTGAGAAA | AGGATATTGCTAGAGTGGAGTCAGAAC | ACCACTGTTATTTGTTCTCACTCCACT | 98 |
| S67 | rs2303754 | C;G | 19:29606361 | 0.44/0.56 | ATGAGAGATAAGCGGGGCCG | CGGACCCATTTCACCCACCA | CCCGACCCTTAACCTCCCC | TGGAGAGGTTGGGACGTTA | 86 |
| S68 | rs11779762 | C;G | 8:104381588 | 0.53/0.47 | GCCTCTGCCATATCCTCAC | AGGTCGGATGTTGGAAAGG | AGACACTTGTGGGACTCAGAAGG | ACAACTGTCTCCTGCTGTCCT | 71 |
| S70 | rs64364409 | C;G | 2:223498844 | 0.46/0.54 | TGGCCCAGTTAGAAGGTGTGGA | CGGCCACCCATCCTGAGAT | ACCCTCCTGTACTGCCGCAC | ACAGTGAAGGTGTGCCCAGT | 97 |
| S77 | rs2072042 | T;C | 16:986640 | 0.40/0.60 | GGGGCTCCAGTTCTGACGAGCGT | GTTTCCGTGAAGTAGCGCT | ATGCTCAGCACACAGGGGA | CACTGCTTCCCCGTGTG | 96 |
| S82 | rs10228737 | T;C | 7:419845 | 0.48/0.52 | TTTGCACTTGACGCACCAGC | CCGAGCAGAGGAAGGAAGTG | TGCAATGAGAGCAGAGGCT | CATCGCAGCCCTCCTGCA | 79 |
| S84 | rs101641176 | T;C | 18:47293478 | 0.54/0.46 | CCTAATCACTCGTGAGGAGTG | ATCCACCATGATGCTCACAA | CCCACGGGAGGAATGTCTTTG | CCCATGGACTTCTGGCC | 71 |
| S94 | rs7072759 | A;G | 10:18282286 | 0.45/0.55 | CTGGGGCAGAGTGGAGAGTC | ATCCACCCTGAACCCAGCC | AGGACACTCGCAGCTGTGG | CAGCGTCCTCTGTGCTACCT | 83 |
| S96 | rs355736520 | T;C | 15:570362495 | 0.49/0.51 | TCCCAGGCTCCAGGTCAGAT | GGATCAATGTGGCTGCTCCCT | TCTCCGCCCTTCTGAGATGC | AGGGCAGAGACTCTGGAACT | 81 |
| S97 | rs8092926 | A;G | 18:85631173 | 0.48/0.52 | AGCCCTGCACACTCACTTACC | TGGCATTCAGATCATCAGGCTTCT | CCATCAGGTGCTGGCACTC | TGCAGGGAAGAGCGCCAG | 83 |
| S105 | rs1265094 | A;G | 6:31139116 | 0.52/0.48 | ACCCCAAGAGGCTTTATAGGGG | CCTTCCCAACGGGTTTGACC | CCACTGGGCTGGCCCCTC | AGTGGAGGAGGGACCAGC | 96 |
| S108 | rs11610836 | C;T | 12:112765162 | 0.55/0.45 | ACACTTCCTGCTGCGTGTCTG | TTCCTCCCCACCACTCCCAT | GGTCCCAGCTGGTCGTGG | ATGCTCCCACACAACCAGCT | 96 |
| S110 | rs13185616 | C;A | 5:137702567 | 0.51/0.49 | GGTCCTACCGAGGTGGGTGA | CATTGCCAAGGACAGAGGGAGA | TTTGGTAGGGAAGGAACTCCCAAT | ATCAGTGGCCATTGTGAGTTCC | 90 |
| S201 | rs2236058 | C;G | 1:12002304 | 0.44/0.56 | GAACACAATAGACAGCTGGTG | CAACACAGGGAATCGTGCTG | TCATCACCCACCTGGTCTG | TCATCACCCGACCTGGTCTG | 83 |
| S202 | rs7536561 | A;G | 1:180274389 | 0.57/0.43 | TCCATATCGTCCCTGCCATC | CGCAATGATGCCCAAATTAC | CCTTTGCTCAATGGGCTGG | CCTTTGCTCAGTGGGCTGG | 81 |
| S203 | rs2224718 | T;C | 1:36599496 | 0.52/0.48 | GAAACGGTGGGAGAAAATTGAAAG | AACTGGACGGGTTCCTGATG | TTGCAGCCAGATACTGGAGAA | TTGCAGCCAGACACTGGAGAA | 87 |
| S204 | rs284001180 | C;T | 2:200005919 | 0.48/0.52 | AGAGCTCAATACCAGATGCTTGG | TAAGGTGGCCATCATTGTTACAG | AGCCGCCACCTCATTCACC | AGCCGCCACTTCATTCACC | 89 |
| S205 | rs285988872 | G;A | 2:200006087 | 0.48/0.52 | CCTGACATGGTGCCTGTTGAC | TTCCAACTCCAGCCTACACAG | CGACCCACGGCCTGCTTC | CGACCCACAGCCTGCTTC | 86 |
| S206 | rs1063353 | A;G | 2:240774229 | 0.49/0.51 | GCGTGCTAAGGGTCTTCATCG | GCCGGTAACTCAAGGACAGC | TGCAGGACTCAAGGCTGCCA | TGCAGGACTCAGGGCTGCCA | 76 |
| S207 | rs2288746 | A;G | 2:240787229 | 0.52/0.48 | TGCCAGCCACACCAGATTC | CCAACGACAACATGTCCTACTCC | AGCGGCCAACGGCAG | AGCGGCCGACGGCAG | 90 |
| S208 | rs2340917 | T;C | 3:14133762 | 0.54/0.46 | GGTGCCCATCTCTGACAGC | CCTGCCAATTTGGACAAAGG | AGTCATTCAATGGCAACAGCC | AGTCATTCACGGCAACAGCC | 90 |
| S209 | rs3213936 | A;G | 5:139137776 | 0.47/0.53 | CCTGGTATTTAGTGGCCATGTC | AACCCTCAAGACGTATTCAGTCC | CCAGCCCTTCAATTGTGGAATC | CCAGCCCTTCGATTGTGGAATC | 74 |
| S210 | rs1801193 | T;C | 5:156344569 | 0.51/0.49 | GGGCCACAGGTATACAAGGTG | AAATCATGAGGACGAGGACAAAG | ATTTATGGCTGGCGGAAACG | ATTTACGGCTGGCGGAAACG | 79 |
| S211 | rs2274514 | C;T | 6:42966762 | 0.51/0.49 | CCCACCCATCTACATCCATTTC | GCACCAGGATCAAGAACTCAGG | CGCCTTTCCGGTGCCCA | CGCCTTTCTGGTGCCCA | 79 |
| S212 | rs592121 | A;G | 6:70274733 | 0.52/0.48 | GCAATTAATGCCAGATGGCTAAC | TAGGGATTAACAGGACCTGATGG | TGTCCCTTTGACCCAATGAG | TGTCCCTTTGGCCCAATGAG | 84 |
| S213 | rs2214326 | G;A | 7:21816533 | 0.52/0.48 | TCCATGTTGACGGATGATGC | CATTCTGTCACTGGGCAGTCC | CAATTGCGCCTGGAATAACG | CAATTGCCACCTGGAATAACG | 66 |
| S214 | rs2293979 | G;A | 8:132625413 | 0.51/0.49 | TTCCAAGTCATCTTCACTGTTGTC | CCTCTTTAGAGACAAAGACCACC | TGTTCCTCTGTGTCTGGTGCCT | TGTTCCTCTATGTCTGGTGCCT | 89 |
| S217 | rs573455 | A;G | 11:117397168 | 0.54/0.46 | TAGCCCTCCGTAGTGCCAAG | ATGCTGCTGGCCAGCTTTC | TTCCTTGTGCAGCAGACACGC | TTCCTTGTGCGCAGACACGC | 89 |
| S218 | rs1044129 | A;G | 15:33866065 | 0.46/0.54 | GGAGCTGCTCGTTTAGGTGAATC | GACCCTGGAGGTATTGGTACG | CCTCAAATACAATGAAGTGCCCA | CCTCAAATACAGTGAAGTGCCCA | 85 |
| S219 | rs169057091 | T;G | 15:42725228 | 0.54/0.46 | ATCTCCATCCGTCCCATCAG | CTCGGTGTGTTTTGCAGTGG | CCACCAGCTCCCGTAGCAAG | CCACCAGCTCCCGGAGCAAG | 82 |
| S220 | rs2684788 | C;T | 15:98961208 | 0.51/0.49 | CAGGGTTTTGTTTCTTCCACACTG | CCGGTAGGTATGTGCACGAG | CCCTTGGAATAACGGCCTCTCC | CCCTTGGAATAATGGCCTCTCC | 74 |
| S221 | rs17715450 | C;A | 16:68695882 | 0.53/0.47 | ACCAACCATCATCCCGACAC | AAGTTGCCGATTTCATCTGG | ACCGTCCTGGCCAGCCA | ACCGTCCTAGGCCAGCCA | 66 |
| S223 | rs2285479 | G;A | 17:10632701 | 0.53/0.47 | CCTCTTCAGCTGCTCGATCTC | CCGAATCCAGCTTGAATTGAC | ATCCTTCTCGGCGATCTTTCTATCA | ATCCTTCTCGGCAATCTTTCTATCA | 88 |
| S226 | rs2229080 | C;G | 18:52906232 | 0.52/0.48 | TCTTGCCCTCTGGAGCATTG | TGGCTGGATTTCGAGCTGAG | AGATCAGCGGACTCCAACCG | AGATCAGCGGGACTCCAACCG | 84 |
| S227 | rs363050 | G;A | 20:10253609 | 0.50/0.50 | TGGGAGGACTCTACATGCTCAGG | TTGCTGAAATGGGACCCTTG | TGTGAATGAGTGGTCGGGCAG | TGTGAATGAGTGATCGGGCAG | 89 |

Assays S43–S110 are taken from Beck et al. [37]; assays S201–S227 are own design. In probes A and B, the SNPs are indicated by bold.

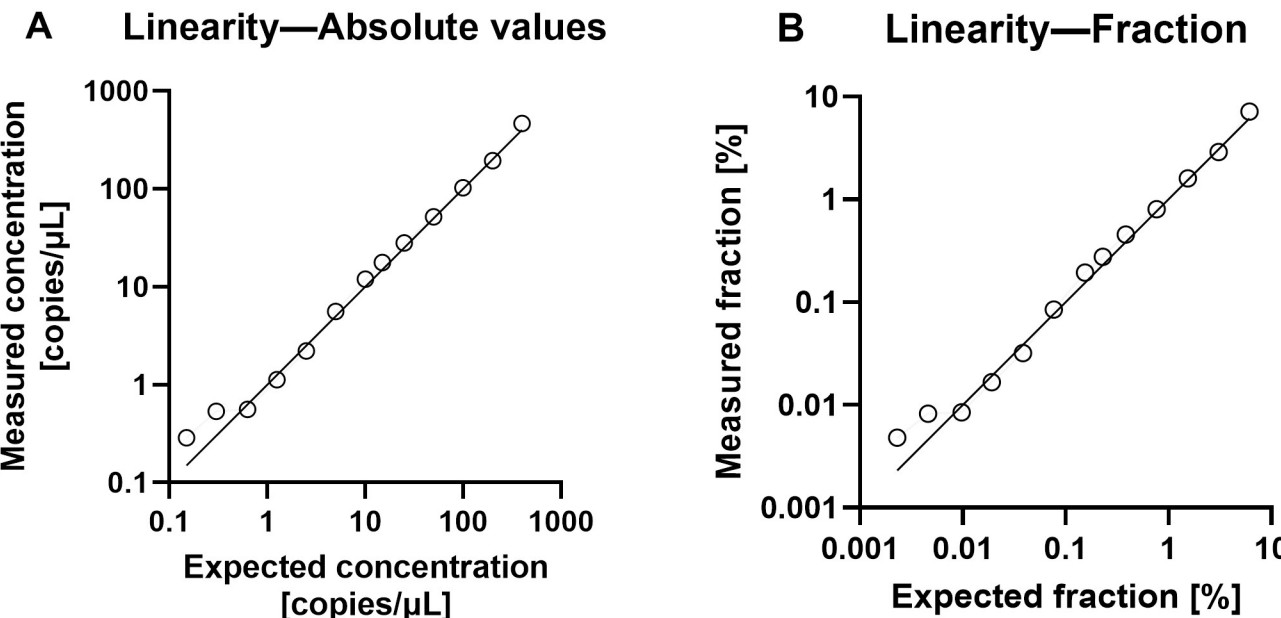

**Fig 1. Linearity.** (A) Showing measured absolute values for DNA concentrations ranging from 0.15 to 400 copies/µL ($n$ = 13). (B) Showing measured fraction for the same values ($n$ = 13). Lines represent identical values. For best view of data points, data are shown on log10 scales. Correlation between measured and expected absolute values: Pearson's r = 0.997 (95%CI = 0.989–0.999). Correlation between measured and expected fractions: Pearson's r = 0.996 (95%CI = 0.986–0.999).

mimicking dd-cfDNA ranged from approximately 10 to 270 copies/reaction in a background ranging from 47,120 to 93,920 copies/reaction. Fractions ranged from 0.02 to 0.37%.

Across Spike 3–5, LOQ with CV<25% of the mimicking dd-cfDNA was estimated to be between 24 and 48 copies/reaction (Spike 3: LOQ = 27–56; Spike 4: LOQ = 24–89; and Spike 5: LOQ<48 copies/reaction). Compiling the data from Spike 3–5, we further estimated the LOQ to be 35 copies/reaction, Fig 4, which was equivalent to 66 copies per mL plasma in our setup using DNA extraction from 4mL plasma.

Lastly, we calculated a 95% confidence interval of the fractions for Spike 3–5, Fig 5. This was used to determine the lowest detectable increase in percentage points for each experiment. All three experiments independently showed that a non-detectable increase at 0.03 percentage points (0.03, 0.031, and 0.034, respectively, for Spike 3–5). The lowest detectable increase in percentage points was 0.04 (0.04, 0.14, and 0.07, respectively, for Spike 3–5), varying because different values were tested. In absolute values, no distinction was possible in measurements with a difference of 8–31 copies/reaction (due to overlapping 95%CIs), whereas a distinction was possible with data where the difference between two points was at least 37 copies/reaction (S1 Fig).

The number of copies per reaction required to identify a certain % increase in the dd-cfDNA fraction was identified theoretically and compared with LOD and LOQ (Fig 6).

## Discussion

We developed a ddPCR-based setup for analyzing dd-cfDNA as a noninvasive marker of organ damage for monitoring organ health in transplanted patients. We tested 40 SNP assays and investigated different essential methodological performance parameters. The ddPCR-based cfDNA quantification method demonstrated high precision, as assessed by comparing expected and measured DNA concentrations. We demonstrated high analytical sensitivity

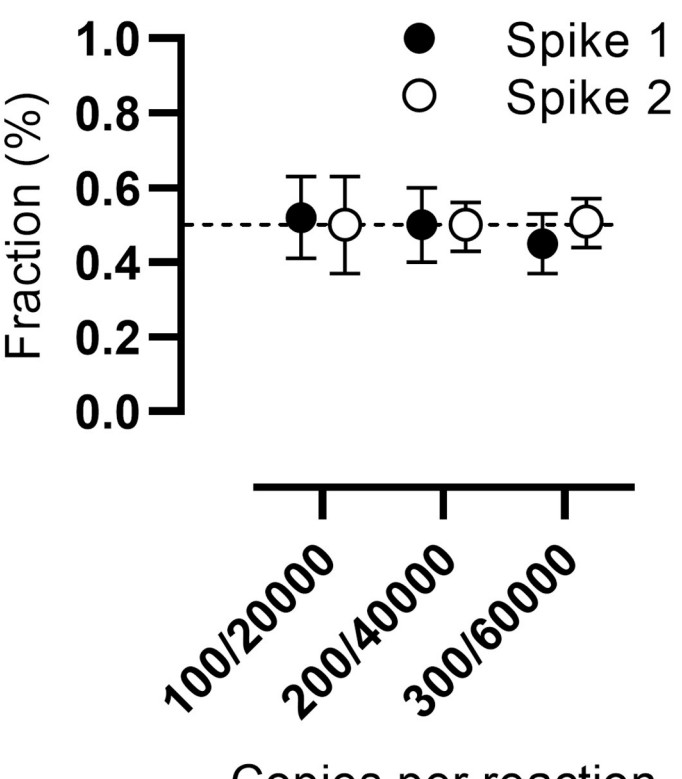

**Fig 2. Measurement of 0.5% cfDNA.** Measurements shown of 0.5% mimicking dd-cfDNA, as constituted by different absolute values of 100, 200, and 300 copies per reaction spiked into backgrounds of 20,000; 30,000; and 60,000 copies per reaction, respectively. Spike 1 represents experiments with probe A, and Spike 2 represents experiments with probe B. Measurements shown as mean fractions. Error bars represent the 95% confidence interval. Dashed line represents 0.5%.

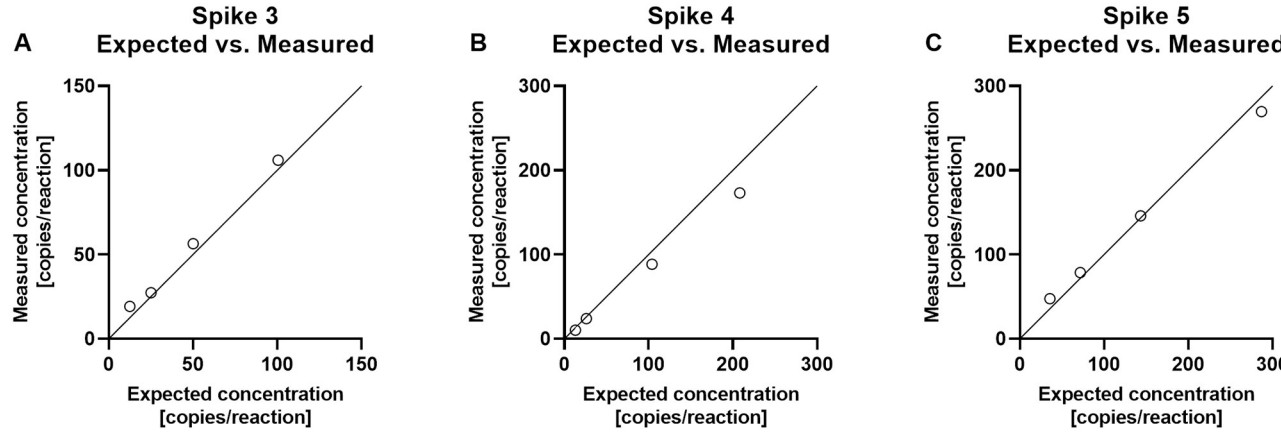

**Fig 3. Linearity of Spike 3–5.** Expected concentration as a function of the measured concentration, both measured in copies/reaction, of three individual experiments, Spike 3–5 (A–C). The diagonal line indicates equal concentrations of expected and measured mimicking dd-cfDNA concentrations. Each dot represents a measured cfDNA concentration. Pearson's r = 0.9987 (95%CI = 0.938–1) for Spike 3, r = 0.9998 (95%CI = 0.990–1) for Spike 4, and r = 0.9998 (95%CI = 0.992–1) for Spike 5.

**Table 2. Results from spike-in experiments 3–5.**

| Experiment | cfDNA | Expected spike-in copies/reaction[a] | Measured copies/ reaction [mean] | CV% of mean | Fraction [%] | CV% of fraction |
|---|---|---|---|---|---|---|
| **Spike 3** | Background | 94,000 | 93,920 | 2.6 | NA[b] | NA |
| | Spike | 12.5 | 19 | 32.2 | 0.02 | 32.7 |
| | Spike | 25 | 27 | 33.8 | 0.03 | 33.7 |
| | Spike | 50 | 56 | 16.5 | 0.06 | 17.0 |
| | Spike | 100 | 106 | 13.2 | 0.11 | 12.4 |
| **Spike 4** | Background | 47,000 | 47,120 | 2.0 | NA | NA |
| | Spike | 13 | 10 | 34.1 | 0.02 | 33.2 |
| | Spike | 26 | 24 | 29.7 | 0.05 | 29.0 |
| | Spike | 104 | 89 | 15.3 | 0.19 | 15.6 |
| | Spike | 208 | 173 | 8.9 | 0.37 | 8.6 |
| **Spike 5** | Background | 94,000 | 93,540 | 3.8 | NA | NA |
| | Spike | 36 | 48 | 17.8 | 0.05 | 17.4 |
| | Spike | 72 | 79 | 14.8 | 0.09 | 15.0 |
| | Spike | 144 | 146 | 10.1 | 0.16 | 10.4 |
| | Spike | 287 | 270 | 9.3 | 0.27 | 9.7 |

[a]Expected spike-in copies/reaction are presented according to recalculation based on the estimated mean concentration of the spike-in material on the ddPCR plate per experiment.
[b]NA = not applicable

with a LOD of 3 copies/reaction, observed at a fraction of 0.002%. This was the lowest DNA concentration that we tested, and it is likely that the actual LOD is lower, presumably 1 copy/ reaction (according to dynamical range of ddPCR and general experience with PCR detection of low levels of DNA [42]). LOQ was determined to be 35 copies/reaction, as defined by a CV of less than 25%.

In dd-cfDNA testing, low dd-cfDNA levels are anticipated; therefore, we decided to study the analytical performance of the method at low absolute levels and at small fractions of cfDNA. We thus studied the detection of fractions of 0.5% and lower. A clear detection of 0.5% was observed in initial studies with acceptable CVs of 7–14%. When examining lower fractions of 0.02–0.37%, we were able to distinguish a difference of 0.04 percentage points as determined by non-overlapping 95% confidence intervals, and we demonstrated that such distinction was only possible with a difference above 0.03 percentage points. These data translated to a difference between two points of >31 copies/reaction, slightly lower than the LOQ of 35 copies/reaction. Further, the LOQ in percentage fraction was estimated to approximately 0.038%.

We based our ddPCR setup on the setup developed by Beck and colleagues [9, 10, 37]. Our setup had some key differences. First, we applied strict *in silico* criteria to the SNP assays, which excluded several of the assays adopted from Beck et al. In our own assay design, we aimed for short amplicons, because short amplicons increase the likelihood for successful amplification of the highly fragmented cfDNA, thereby increasing assay sensitivity [8], which is a generally known issue in cfDNA testing [43]. In addition, we used a different preamplification system using a targeted approach rather than a universal approach with whole-genome-amplification [9].

Overall, the methodological performance of our ddPCR setup was acceptable and in line with reports from other validation studies. Beck and colleagues reported a LOD estimated at about 10 copies per mL plasma corresponding to a fraction of 0.15% dd-cfDNA [10]. This is in

# Estimation of LOQ

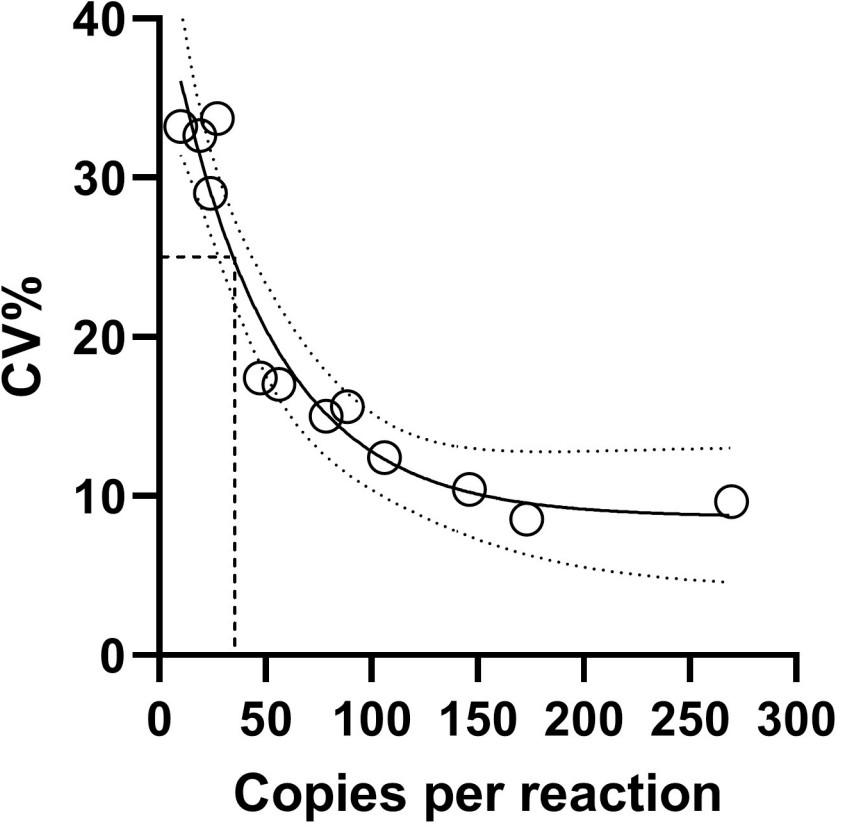

**Fig 4. Estimation of LOQ.** Non-linear curve fitting of data points from Table 2 from Spike 3–5. Goodness of fit $R^2 =$ 0.935; fit indicated by line and 95% confidence interval indicated by dotted lines. According to curve equation, a CV<25% was reached at 35 copies per reaction, indicated on the figure by dashed lines.

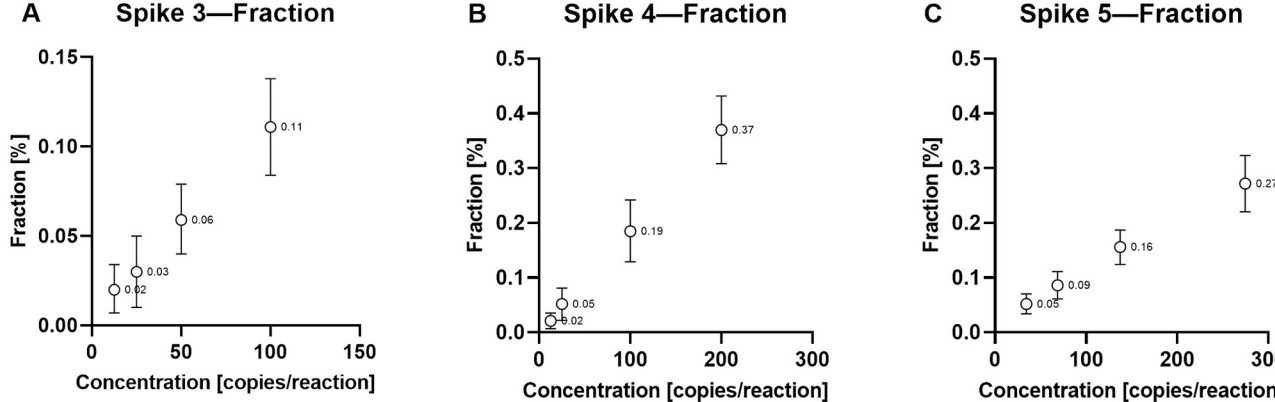

**Fig 5. Estimation of minimal detectable percentage point for Spike 3–5.** Three independent experiments, Spike 3–5 (A–C). The x-axis of each plot shows the expected cfDNA concentration of the mimicking dd-cfDNA in copies/reaction, while the y-axis shows the calculated fraction of the mimicking dd-cfDNA in percentage. Each dot represents the mean value of the fraction for a measured concentration (data value labeled), while the error bars represent the 95% confidence interval of the mean fraction. Minimal detectable percentage points were estimated between non-overlapping confidence intervals (Spike 1: 0.06%–0.02% = 0.04%; Spike 2: 0.19%–0.05% = 0.14%; Spike 3: 0.16%–0.09% = 0.07%).

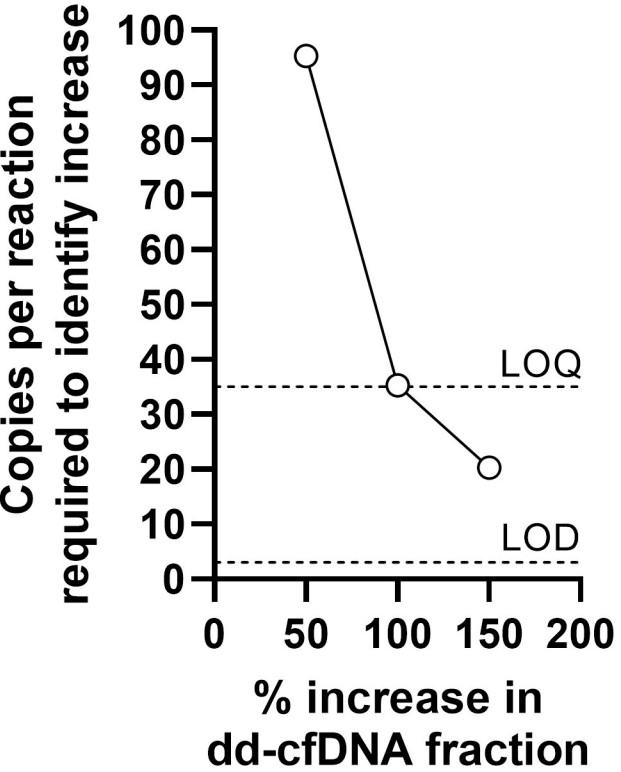

**Fig 6. Percentage increase compared to LOD and LOQ.** Theoretically estimated minimal copy numbers required to detect a certain percentage increase of dd-cfDNA fraction (50%, 100% and 150%), compared to LOD and LOQ (represented by dashed lines).

line with our LOD being equivalent to 6 extractable copies per mL plasma, although our fraction was 0.002%. Beck et al. also showed a repeated testing of samples spiked with 2% DNA yielding CVs of 4–14% [9]. This is also in line with our CVs of 9–11% at a fraction of 0.5%.

Also using ddPCR, Magnussen and colleagues presented highly sensitive and precise measurements of spiked samples mimicking dd-cfDNA from 0.005% to 1% [31]. They reported a LOD of 0.055% [31].

The commercially available NGS-based AlloSure test system reported a LOD of 0.16% and a LOQ of 0.22% in their original validation study (LOQ set at CV<20%) [44]. Later, new performance data were presented for an updated version, AlloSure 3.0, where the LOD now was 0.12%, and with reduced CVs, reported to be well below 10% throughout the range for decision making [45]. Despite higher LODs and LOQ from AlloSure when compared as fractions, the LOD data from AlloSure are comparable to our data when recalculated to estimates of absolute values. The CVs from AlloSure are lower, however, which is likely due to the higher number of SNP targets used in their system. The reportable range is from 0.12% to 16% [45], in which the upper limit is due to the technical and statistical approach for calling SNPs, likely interfering with heterozygosity in the event of high dd-cfDNA levels [8].

Using spike experiments with synthetic DNA (gBlocks), Kokelj et al. analyzed fractions down to 0.1% using ddPCR [39]. Their LOQ was 10–20 copies/reaction (defined by a CV<20%) [39]. In our study we estimated the LOQ to approximately 35 copies/reaction, with a CV<25% (we chose an acceptable CV at <25%, as recommended elsewhere [46, 47]). These different LOQs may reflect the difference in test material, where we used cfDNA from plasma (assumed to be highly similar to dd-cfDNA concerning fragmentation and other characteristics). We did detect a fraction of 0.1% at a CV of 12.4%, and we managed to analyze fractions as low as 0.052% without having a CV>20%.

Importantly, a fraction is determined by two underlying values. Consequently, changes in hidden absolute cfDNA values may impact the dd-cfDNA fraction measurement, so that it is not always representative of the underlying events which can lead to misinterpretations or a less accurate picture of the clinical status of the organ. This was recently demonstrated with ddPCR testing of lung transplantations, where a decrease in dd-cfDNA fraction gave the wrong impression that the level of dd-cfDNA was decreasing whereas absolute cfDNA measurements provided the actual insight into the organ status [31].

The LOQ reflects the certainty of the quantitative measurement. However, for the purpose of identifying a potential increase in dd-cfDNA fraction, absolute certainty of the quantitative measurement is not necessary. The important parameter is the capability of distinction between truly different values versus stochastic or biological variation. With our ddPCR-based method, we found a remarkable capability to identify an increase of only 0.04 percentage points, observed at small fractions. An important aspect, relevant for long-term monitoring of patients, is how a specific increase in percentage relates to the indication of organ damage or risk of later rejection (as opposed to using a threshold). For example, as with a 100% increase, where a fraction would increase, for example, from 1% to 2%. We estimated that an increase of 100% was identifiable around the same value as our absolute LOQ, whereas an increase in 50% was identifiable only when absolute cfDNA copies were close to 100 copies per reaction. An increase of 150% would be easy to identify. Despite that these values are based on theoretical calculations they do indicate sufficient capabilities for such a way forward for an algorithm to judge clinical data. Interestingly, it was recently observed with kidney transplantations that an approximate 150% increase between two subsequent samples was indicative of graft injury [5]. Future clinical studies are needed to identify applicable thresholds or difference between subsequent samples—to confirm the best algorithm for indication of organ damage.

Our study had some study limitations. First, LOD and LOQ was only studied for a selected number of assays and not all assays. However, we believe that the observed data can be viewed as exemplary for all assays. Also, we studied spiked-in samples which may not comprise all the challenges and variation observed in clinical samples. However, it is not uncommon to use such mimicking samples for methodological assessments [39, 44], because the use of spike-in experiments allows for an easily controllable experiment, where cfDNA concentrations and fractions are adjustable [39]. Thus, we used these experiments to assess key analytical performance parameters of ddPCR relevant for using this method for quantitative measurement of dd-cfDNA in transplantation patients. Future studies will look at the clinical application and the clinical value of this method.

In general, the different molecular techniques for testing dd-cfDNA are highly sensitive. ddPCR offers higher precision and better analytical specificity than qPCR [35, 38]. NGS may offer higher precision due to analyzing more SNP targets than with ddPCR [44]. Compared to NGS, ddPCR is often said to have a shorter turnaround time [9, 27, 34], and to be more cost effective [34, 36], which makes ddPCR more practical for routine use [34, 38]. Importantly, ddPCR also allows for quantification of absolute values and not only fractions, which gives additional important information for patient evaluation in a clinical setting. ddPCR offers an

excellent monitoring tool, providing a reliable method for repeated testing of a patient over time; once the informative SNPs are identified, any additional analysis is simple, cost efficient, and rapid. We have thus implemented a method which is ready to assist clinicians in the monitoring and management of transplantation patients, providing a noninvasive monitoring marker indicative of organ damage and thus organ health, which may contribute to prolonging organ and thus patient survival. Additional patient groups may benefit from this method. In principle, any chimeric situation may benefit from this method. Also, numerous research activities as well as different types of cellular treatment may benefit from using this method.

## Conclusion

The results from this study showed that our ddPCR method is highly sensitive, precise, and allows for quantitative assessment of low levels of cfDNA. This analytical validation of 40 SNP assays showed that the method is applicable for testing and monitoring of dd-cfDNA in all transplantation patients providing a noninvasive marker for organ damage. Future studies will investigate the clinical value of dd-cfDNA testing in transplantation patients.

## Supporting information

**S1 Text. SNP assay design criteria.**
(DOCX)

**S1 Fig. Delta copies of mimicking dd-cfDNA versus percentage increase in fraction.** Delta copies of data from Spike 3–5 plotted against percentage increase of fraction shown to mark a difference (dashed line of 37 copies per reaction) between distinguishable and non-distinguishable data.
(TIF)

## Acknowledgments

We thank Birgitte Bundgaard and her staff in the Laboratory of Blood Genetics, at the Department of Clinical Immunology, Copenhagen University Hospital.

## Author Contributions

**Conceptualization:** Frederik Banch Clausen, Grethe Risum Krog.

**Data curation:** Frederik Banch Clausen, Kristine Mathilde Clara Lund Jørgensen, Lasse Witt Wardil, Grethe Risum Krog.

**Formal analysis:** Frederik Banch Clausen, Kristine Mathilde Clara Lund Jørgensen, Lasse Witt Wardil, Leif Kofoed Nielsen, Grethe Risum Krog.

**Funding acquisition:** Frederik Banch Clausen, Grethe Risum Krog.

**Investigation:** Frederik Banch Clausen, Kristine Mathilde Clara Lund Jørgensen, Lasse Witt Wardil, Grethe Risum Krog.

**Methodology:** Frederik Banch Clausen, Kristine Mathilde Clara Lund Jørgensen, Lasse Witt Wardil, Leif Kofoed Nielsen, Grethe Risum Krog.

**Project administration:** Frederik Banch Clausen, Leif Kofoed Nielsen, Grethe Risum Krog.

**Resources:** Frederik Banch Clausen, Leif Kofoed Nielsen, Grethe Risum Krog.

**Supervision:** Frederik Banch Clausen, Leif Kofoed Nielsen, Grethe Risum Krog.

**Validation:** Frederik Banch Clausen, Kristine Mathilde Clara Lund Jørgensen, Lasse Witt Wardil, Grethe Risum Krog.

**Visualization:** Frederik Banch Clausen, Kristine Mathilde Clara Lund Jørgensen, Lasse Witt Wardil, Grethe Risum Krog.

**Writing – original draft:** Frederik Banch Clausen, Kristine Mathilde Clara Lund Jørgensen, Lasse Witt Wardil.

**Writing – review & editing:** Frederik Banch Clausen, Kristine Mathilde Clara Lund Jørgensen, Lasse Witt Wardil, Leif Kofoed Nielsen, Grethe Risum Krog.

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
