## [Decision Letter · Decision Letter 0]

4 Jan 2023

PONE-D-22-34954Droplet Digital PCR-based testing for donor-derived cell-free DNA in transplanted patients as noninvasive marker of allograft health: Methodological aspectsPLOS ONE

Dear Dr. Clausen,

Thank you for submitting your manuscript to PLOS ONE. After careful consideration, we feel that it has merit but does not fully meet PLOS ONE’s publication criteria as it currently stands. Therefore, we invite you to submit a revised version of the manuscript that addresses the points raised during the review process.

We look forward to receiving your revised manuscript.

Kind regards,

Benjamin M. Liu, MBBS, PhD, D(ABMM), MB(ASCP)

Academic Editor

PLOS ONE

Journal Requirements:

Additional Editor Comments:

Although reviewers suggested this is an technically sound study, the authors did not validate their assay in patients with organ transplantation. While analytical validation is necessary and using mimic samples is considered fine for this, clinical validation using a small number of samples from organ transplantation patients will be expected and encouraged.

Reviewers' comments:

Reviewer's Responses to Questions

**Comments to the Author**

1. Is the manuscript technically sound, and do the data support the conclusions?

Reviewer #1: Yes

2. Has the statistical analysis been performed appropriately and rigorously? 

Reviewer #1: Yes

3. Have the authors made all data underlying the findings in their manuscript fully available?

Reviewer #1: Yes

4. Is the manuscript presented in an intelligible fashion and written in standard English?

Reviewer #1: Yes

5. Review Comments to the Author

Reviewer #1: The authors describe the application of digital droplet PCR to address the relevant clinical issue of allograft health. The technique is powerful and the methodological aspects investigated are appropriate and comprehensive. In my opinion, statistical analysis is appropriate and results presentation is clear.

I have no specific comments for the authors.

6. PLOS authors have the option to publish the peer review history of their article (what does this mean?). If published, this will include your full peer review and any attached files.

Reviewer #1: **Yes: **Maria Lorena Abate

---

## [Author Response · Author response to Decision Letter 0]

8 Feb 2023

Response to Reviewers

I thank the editor and the reviewer for their work and positive comments regarding our manuscript. It is highly appreciated. A new and revised manuscript has been uploaded, with and without track changes as requested by the editor.

Regarding Journal Requirements: 

Ad 1) I have ensured to the best of my knowledge that the manuscript meets the PLOS ONE’s style requirements, which includes a change in the manuscript reference style for supplementary information. The specific reference reading “…in supporting information S1 Text.” was changed to “…in S1 Text.” (Page 7). I hope this is now correct.

Ad 2) I have matched the information provided in the ‘Funding Information’ and ‘Financial Disclosure’. (I understand ‘Financial Disclosure’ as the comment of financial interest which is stated in the Cover Letter). Consequently, this new comment in the new Cover Letter should be viewed as the information also for Funding Disclosure’. In addition, I have provided grant numbers for each grant, as requested. 

Ad 3) An ethics statement was added as the first section under Meth-ods (page 5 and 6). It reads as follows:

“This study was a quality assurance project using blood samples from healthy volunteers. Informed and written consent was obtained from all participating volunteers when blood samples were collected. According to Danish law, no ethical approval was required, because this was a quality assurance project, thus waiving the need for ethics committee approval of the study. No DNA information related to disease was examined, and only SNPs with no known clinical significance were used, thus avoiding the challenges of reporting incidental findings.”

Regarding additional Editor Comments:

On the issue of including a small validation of clinical samples, I agree that one would often expect to see an analytical validation accompanied by a clinical validation. And adding a few clinical samples might seem like a good idea and a natural step towards a clinical validation. We did exactly that in our first study on cell-free DNA, almost 20 years ago [PMID: 16231312]. However, this field is far more complicated. Thus, for a number of key reasons, as listed below, adding a small number of clinical samples will not improve our manuscript. And we therefore choose not to.

1. A true clinical validation is much larger than often realized. A clinical validation in the context of transplantation patients is complicated. Importantly, no actual clinical validation has yet been done in this field. Such validation must include a) monitoring several patients over several years just to demonstrate correlation; b) a potential threshold must be identified and tested as the indication for action and medical intervention; c) a threshold-based guidance of medical intervention must then be tested prospectively to demonstrate clinical value. We mention these important issues in our manuscript. A clinical valida-tion is thus unfeasible at this time. Adding a small validation should not be regarded as a clinical validation.

2. A small number of clinical samples is unnecessary. The only reason that you would add a number of clinical samples to a methodological study is when your mimicking material is vastly different from the clinical material. For example, several studies have used artificial DNA samples and then clinical plasma samples to see if the presented assays would work in plasma as well. We used plasma samples obtained and processed under the same clinical conditions as expected for clinical plasma samples. The cfDNA in plasma is similar. Similar in fragmentation and size. We discuss these issues in the manuscript. Thus, adding a small validation would add no real value.

3. Meticulous presentation of the methodological aspects is crucial. In transplantation, cfDNA testing is a young science, and it is difficult to convey important methodologically details to clinicians and medical scientists. Therefore, special attention to methodological aspects is needed and warrants meticulous presentation in the literature. The issue of threshold in kidney testing is a good example of confusion related to methodological aspects and insufficient validation. No precise threshold has been identified to guide medical intervention; however, many clinicians believe that such thresholds exist—even though the most recent paper clearly shows otherwise [PMID: 34953773]. We discuss these issues in the manuscript. Important methodological issues are thus not always sufficiently dealt with. Adding a small clinical validation may only add further confusion about assay readiness. By contrast, a clear separation between analytical and clinical validation is necessary and desirable in my opinion. Focus should be on the methodological aspects at first. 

4. Timely publication of method is highly important. As discussed under point 1, a clinical validation requires several steps and monitoring of patients over several years. Therefore, it is important to present the method for other scientists quickly in the literature to help others start similar endeavors to save patient lives. The ddPCR assay we present is an older method (taken from the literature) that we have improved significantly. We could only re-use half of the assays from the old method and thus designed new assays. This is a very important message to get out quickly, so that other scientists will not waste unnecessary resources trying to implement the old setup. As such, the key information in our manuscript is to present reliable SNP assays which can be used for dd-cfDNA testing in plasma.

5. In addition, ethical rules restrict us from ad hoc testing a small number of samples. Ethics committee rules imply that we cannot just add a small group of samples. To test clinical samples, such samples must be part of a clinical project, in which a sufficient sample size must be calculated and accounted for, including an accepted and funded feasibility plan to obtain clinical goals. GDPR rules prevent us from just adding a small group of samples, if tested outside an approved clinical study. So, even if we agreed that this was the correct way forward, we would not be allowed to do so. 

Based on these five key arguments, we abstain from adding clinical samples. Rather, we adhere to the rationale of presenting methodologically aspects timely, and then present thorough clinical data in years to come. Thus, we now focus our resources on planning differ-ent clinical projects in which we will apply this method. In the mean-time, it is important to publish this method and our meticulous description.

Thank you for your patience reading our arguments for this position.

I hope the editor will approve of these considerations.

And I thank again the reviewer for the reviewer’s highly positive feedback.

---

## [Editor Report · Decision Letter 1]

9 Feb 2023

PONE-D-22-34954R1Droplet Digital PCR-based testing for donor-derived cell-free DNA in transplanted patients as noninvasive marker of allograft health: Methodological aspectsPLOS ONE

Dear Dr. Clausen,

Thank you for submitting your manuscript to PLOS ONE. After careful consideration, we feel that it has merit but does not fully meet PLOS ONE’s publication criteria as it currently stands. Therefore, we invite you to submit a revised version of the manuscript that addresses the points raised during the review process.

We look forward to receiving your revised manuscript.

Kind regards,

Benjamin M. Liu, MBBS, PhD, D(ABMM), MB(ASCP)

Academic Editor

PLOS ONE

Journal Requirements:

Additional Editor Comments (if provided):

Please update the format of result section. Fig title should not be subtitle of result section. Sub-section of result should be results from one or more one figures. Please check published PLOS One and use this paper as an example to improve your manuscript: https://journals.plos.org/plosone/article?id=10.1371/journal.pone.0159729

---

## [Author Response · Author response to Decision Letter 1]

11 Feb 2023

Response to Reviewers

I thank again the editor for the editor’s work on improving our manuscript.

Regarding Journal Requirements

I have reviewed all the references. None of them cite papers that have been retracted.

However, references 40 and 45 do not appear at PubMed. 

Reference 40 is an abstract, and it is still available and can be found on google scholar. 

Reference 45 is a white paper from a commercial company, and it can also be found on google scholar. As all cited papers are available, I have made no amendments to the reference list.

Regarding Additional Editor Comments

This issue with the format of the result section is based on a simple mistake. 

The fig titles are not sub-sections of the result section (the result section has no sub-sections). It is the caption and legend of each figure, inserted into the text after the paragraph where the figure is first mentioned. This has been done according to the guidelines from PLOS ONE. Specifically, the guidelines state: “Figure captions are inserted immediately after the first paragraph in which the figure is cited.” Elsewhere it says: “The caption may also include a legend as needed.” Then, one can download an example (Download sample manuscript body), in which the fig caption and legend is presented exactly as I have done it in the manuscript. Thus, I have followed the journal’s instructions meticulously. I do agree that this insertion may be confusing. To alleviate the confusion, I have added the following sentence on top of each fig paragraph: “[The paragraph below is Fig 1’s caption and legend]” (writing Fig 1’s for Fig 1 and Fig 2’s for Fig 2 and so forth). In this way, one should deduce that the paragraph is not a part of the text of the result section.

I hope this is helpful, although it is a deviation from the guidelines.

---

## [Editor Report · Decision Letter 2]

14 Feb 2023

Droplet Digital PCR-based testing for donor-derived cell-free DNA in transplanted patients as noninvasive marker of allograft health: Methodological aspects

PONE-D-22-34954R2

Dear Dr. Clausen,

We’re pleased to inform you that your manuscript has been judged scientifically suitable for publication and will be formally accepted for publication once it meets all outstanding technical requirements.

Kind regards,

Benjamin M. Liu, MBBS, PhD, D(ABMM), MB(ASCP)

Academic Editor

PLOS ONE
---

## [Editor Report · Acceptance letter]

16 Feb 2023

PONE-D-22-34954R2 

Droplet Digital PCR-based testing for donor-derived cell-free DNA in transplanted patients as noninvasive marker of allograft health: Methodological aspects. 

Dear Dr. Clausen:

I'm pleased to inform you that your manuscript has been deemed suitable for publication in PLOS ONE. Congratulations! Your manuscript is now with our production department. 

Kind regards, 

on behalf of

Dr. Benjamin M. Liu 

Academic Editor

PLOS ONE